# Evaluation of the FluorWPS Model and Study of the Parameter Sensitivity for Simulating Solar-Induced Chlorophyll Fluorescence

Chiming Tong [1], Yunfei Bao [1,*], Feng Zhao [2], Chongrui Fan [2], Zhenjiang Li [2] and Qiaolin Huang [1]

[1] Beijing Institute of Space Mechanics and Electricity, China Academy of Space Technology, Beijing 100094, China; tongchiming.1989@163.com (C.T.); hqiaolin_vip@sina.com (Q.H.)
[2] School of Instrumentation Science and Opto-Electronics Engineering, Beihang University, Beijing 100191, China; zhaofeng@buaa.edu.cn (F.Z.); fanchongrui@buaa.edu.cn (C.F.); Lizhenjiang@buaa.edu.cn (Z.L.)
* Correspondence: byf_rs@163.com

**Abstract:** Solar-induced chlorophyll fluorescence (SIF) has been used as an indicator for the photosynthetic activity of vegetation at regional and global scales. Canopy structure affects the radiative transfer process of SIF within canopy and causes the angular-dependencies of SIF. A common solution for interpreting these effects is the use of physically-based radiative transfer models. As a first step, a comprehensive evaluation of the three-dimensional (3D) radiative transfers is needed using ground truth biological and hyperspectral remote sensing measurements. Due to the complexity of forest modeling, few studies have systematically investigated the effect of canopy structural factors and sun-target-viewing geometry on SIF. In this study, we evaluated the capability of the Fluorescence model with the Weighted Photon Spread method (FluorWPS) to simulate at-sensor radiance and SIF at the top of canopy, and identified the influence of the canopy structural factors and sun-target-viewing geometry on the magnitude and directional response of SIF in deciduous forests. To evaluate the model, a 3D forest scene was first constructed from Goddard's LiDAR Hyperspectral and Thermal (G-LiHT) LiDAR data. The reliability of the reconstructed scene was confirmed by comparing the calculated leaf area index with the measured ones from the scene, which resulted in a relative error of 3.5%. Then, the performance of FluorWPS was evaluated by comparing the simulated at-sensor radiance spectra with the spectra measured from the DUAL and FLUO spectrometer of HyPlant. The radiance spectra simulated by FluorWPS agreed well with the measured spectra by the two high-performance imaging spectrometers, with a coefficient of determination ($R^2$) of 0.998 and 0.926, respectively. SIF simulated by the FluorWPS model agreed well with the values of the DART model. Furthermore, a sensitivity analysis was conducted to assess the effect of the canopy structural parameters and sun-target-viewing geometry on SIF. The maximum difference of the total SIF can be as large as 45% and 47% at the wavelengths of 685 nm and 740 nm for different foliage area volume densities (FAVDs), and 48% and 46% for fractional vegetation covers (FVCs), respectively. Leaf angle distribution has a markedly influence on the magnitude of SIF, with a ratio of emission part to SIF range from 0.48 to 0.72. SIF from the grass layer under the tree contributed 10%+ more to the top of canopy SIF even for a dense forest canopy (FAVD = 3.5 m⁻¹, FVC = 76%). The red SIF at the wavelength of 685 nm had a similar shape to the far-red SIF at a wavelength of 740 nm but with higher variability in varying illumination conditions. The integration of the FluorWPS model and LiDAR modeling can greatly improve the interpretation of SIF at different scales and angular configurations.

**Keywords:** solar-induced chlorophyll fluorescence (SIF); deciduous forests; Fluorescence model with weighted photon spread method (FluorWPS); 3D canopy structure; G-LiHT LiDAR

## 1. Introduction

Photosynthesis is an important process for a terrestrial ecosystem. Vegetation captures photons, which then dissipated in three pathways, including driving photosynthesis, dissipating as heat and being emitted as sun-induced chlorophyll fluorescence (SIF) [1]. SIF, ranging from 640–850 nm, is emitted by Photosystems I (PSI) and II (PSII). It is characterized by two peaks at 685 nm and 740 nm, respectively. The Fraunhofer line discriminator (FLD) principle is used to decouple the SIF from the reflected radiance. SIF is closely related to photosynthesis, so it can be considered as a direct indicator of the functional status of photosynthetic machinery [2]. SIF has been used to monitor phenology [3,4], detect plant stress [5–7] and estimate gross primary productivity (GPP) [8–11], among many other applications. Substantial advances have been made in the retrieval of SIF from ground [12,13], airborne [14,15] and satellite measurements [16–18].

Canopy structure is a crucial factor for impacting the relationship between SIF and photosynthesis [19]. It is expected that the canopy structure can affect the light distribution within the canopy, and thereby, photosynthesis and fluorescence emission of leaves. Photons in the absorbed photosynthetic active radiation (APAR) range are absorbed by chlorophyll molecules in the leaf, then re-emitted as fluorescence photons. Fluorescence is determined by the fluorescence emission efficiency and the intensity of absorbed energy [20]. When light penetrates into the deeper canopy layer, the canopy structure affects the light quality and intensity [21]. The canopy structure also influences the re-absorbed and scattering processes of SIF. Within the canopy, SIF at 685 nm (red SIF) is re-absorbed more and scattered less than that at 740 nm (far-red SIF) [22]. The far-red SIF experiences further scattering with canopy elements, and therefore, the scattered portion reaching the sensor is higher than that of red SIF [23]. The scattering effect of the far-red SIF is sensitive to the leaf angle inclination, clumping of leaves, leaf size and amount of leaf area [24,25]. Therefore, SIF at the top of canopy (TOC) depends on the incident light distribution and the radiative transfer process. Furthermore, similar to the directional effects of vegetation reflectance, SIF varies with changes in sun-target-viewing geometry [26,27]. As a result, it is important to understand the radiative transfer process of SIF within the canopy if we want to establish a quantitative link between photosynthesis and SIF.

Radiative transfer models (RTMs) can simulate the distribution of light within the canopy and investigate the effects of canopy structure on the physical processes of the absorption and scattering of SIF. Among them, the Soil-Canopy Observation of Photochemistry and Energy (SCOPE) model has been widely used for simulating SIF for one-dimensional (1D) continuous canopies [28]. SCOPE is highly efficient due to the analytical computation for radiative transfer of the incident fluxes. However, it is not appropriate for assessing the impact of three-dimensional (3D) structural heterogeneity both in horizontal and vertical dimensions [29–31]. Modeling of the heterogeneity of terrestrial vegetation (e.g., forest) needs 3D radiative transfer models. The fluorescence model with weighted photon spread method (FluorWPS) based on the theory of the Monte Carlo ray-tracing can simulate the scattering and absorption of fluorescence for various canopy structures [32]. The FluorWPS uses a number of four-sided polygons or triangular or disks to represent the canopy. The accuracy of FluorWPS was demonstrated with the comparison of the SCOPE model for 1D canopies and further evaluated by the reconstructed SIF spectra from measurements of row-structured canopy. Based on the Monte Carlo ray tracing method, the FluorFLIGHT model calculates the reflectance and fluorescence at the TOC level. Tree crowns are modeled by a set of geometric primitives filled with turbid media, such as ellipsoidal and conical geometries, and a simple growth model is used to limit the overlap between neighboring crowns [33]. Compared with the airborne hyperspectral imagery, there were some deviations between absolute SIF values retrieved with 6.5 nm full width at half maximum (FWHM) and simulated at 1 nm resolution [34]. The modified version of FluorFLIGHT can simulate the SIF emitted from the understory and was validated with airborne hyperspectral data [35]. A fluorescence module was added

in the Discrete Anisotropic Radiative Transfer (DART) model by coupling with the Fluspect-B model to simulate TOC SIF. DART uses planar elements to simulate forests architecture accounting for the leaf clumping at branch level [36]. The simulation accuracy has been evaluated through a comparison with the measured reflectance, but has not been compared with the SIF radiance. The sensitivity analysis of DART revealed that canopy structural parameters drove the intensity and spectral characteristics of TOC SIF in the nadir direction in a birch forest [37]. The Fluorescence Radiative Transfer model based on Escape and Recollision probability (FluorRTER) relies on spectral invariants properties to simulate the SIF of 3D heterogeneous canopies from airborne or satellite measurements [38]. The accuracy of the FluorRTER model was evaluated with SCOPE and the FluorWPS model. Although the influences of the solar zenith angle (SZA) and canopy structure parameters on TOC SIF were analyzed in FluorRTER, the understory layer was not considered. Previous studies have validated the capability of 3D RTMs and examined the influence of canopy structural parameters on TOC SIF. However, there is no comprehensive evaluation work for 3D RTMs using airborne hyperspectral remote sensing measurements and concurrent in situ measurements, especially for complex structural canopy. In addition, few studies systematically investigated the effect of canopy structural factors and sun-target-viewing geometry on TOC SIF.

The newly FluorWPS model presents some features that make it possible to simulate radiance considering the atmospheric contribution, whereas this capability has not been evaluated [39]. In this study, we aim to evaluate the FluorWPS model with airborne hyperspectral measurements and the DART model, and perform a sensitive analysis of its input parameters. The specific objectives are: (1) to validate the capability of the FluorWPS model to simulate at-sensor hyperspectral radiance received by an airborne sensor; (2) to compare the TOC SIF value simulated by the FluorWPS model with that of the DART model; (3) to identify the influence of the canopy structural factors and sun-target-viewing geometry on the magnitude and directional response of SIF in deciduous forests. For these objectives, the FluorWPS model will be used and the simulated radiance spectra are tested using the data measured from two high-performance imaging spectrometers of HyPlant. Moreover, we compared TOC SIF simulated by the FluorWPS model and the DART model. Sensitivity analysis will be conducted to determine the potential of FluorWPS for interpreting the influence factors of SIF.

## 2. Materials and Methods

### 2.1. Description of the Study Site

This study was conducted in the Duke Forest located in central North Carolina (Lat 35°58′25.0″N, Lon 79°6′1.4″W). The altitude is in the range of 91–232 m and the area is over 2800 hm². The Duke Forest includes mixed deciduous forest and loblolly pine stands. Hickory and oak are the main dominant tree species in deciduous forest. The height of the canopy is in the range of 2–35 m in 2013. The forest age within the study site is in the range of 3–46 years old. There are numerous large gaps between the canopies. The mean annual temperature is 15.5 °C and the annual precipitation is about 1100 mm uniformly distributed across seasons [40]. The forest was significantly experienced with natural disturbances, such as a drought from the late growing seasons of 2001 to October 2002. We selected a squared region of interest (60 m × 60 m) near the hardwood tower as the study area (Figure 1).

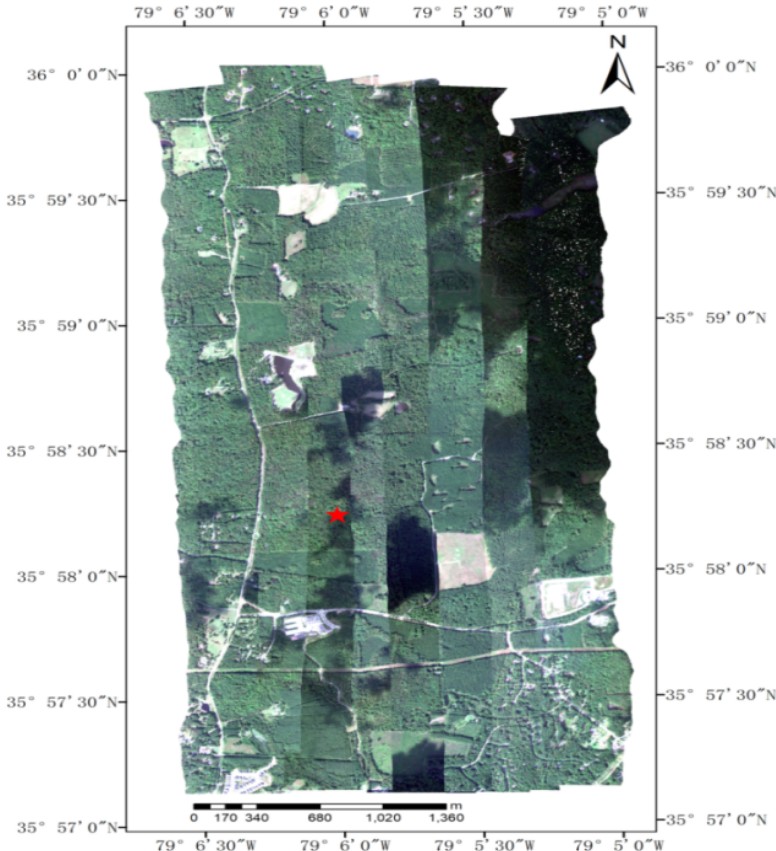

**Figure 1.** A Red-Green-Blue (RGB) composite map of the Duke Forest was combined using reflectance bands at 657.4 nm, 564.4 nm and 484.7 nm for the red, green and blue band, respectively. The red star is the hardwood tower.

### 2.2. Datasets for Model Evaluation

The accuracy of the FluorWPS model was firstly evaluated with the airborne hyperspectral measurements acquired in the FLEX-US campaign, which was conducted in 2013 in North Carolina, USA. The aim of the campaign was to study ecosystem health and carbon cycle dynamics of vegetation. Both the European HyPlant imaging spectrometer systems and the NASA operated G-LiHT LiDAR systems were mounted on a King Air aircraft [41]. The campaign was divided into two main time windows. The first main campaign window started from 26 September to 7 October. The second campaign window was from 24 to 27 October. The campaign delivered a data set of hyperspectral, thermal and LiDAR data, as well as intensive ground and leaf level measurements. With canopy structural parameters (e.g., crown height, crown width, height to live crown base) retrieved from G-LiHT LiDAR data and concurrent field measurements (e.g., leaf reflectance and transmittance), FluorWPS was implemented to simulate at-sensor radiance spectra. The simulated spectra were then compared with the measured spectra of HyPlant. Detailed collected and processing methods for the measurements are explained below.

The accuracy of SIF retrieval from airborne measurements is affected significantly by the atmospheric and lighting conditions as well as the characterization of the instrument. The accuracy of retrievals from the HyPlant in the study area cannot be guaranteed due to a lack of reference retrievals by TOC SIF. Therefore, we further compared the accuracy of SIF simulated by the FluorWPS model with that of the DART model.

### 2.2.1. Ground Measurements

Ground-based data were collected during the first main campaign window. Some ground-measuring facilities were gathered to create a network to record leaf level and ground measurements during the campaign windows. An ASD spectrometer (FieldSpec 3, Analytical Spectral Devices, Inc., Bolder, Co., USA) equipped with an external integrating sphere (LI-1800, Li-Cor, Lincoln, NE, USA) was used to measure tree leaf reflectance and transmittance spectra. Tree leaf fluorescence was measured by the custom FluoWat leaf clip coupled with a portable FieldSpec FR spectrometer (ASD, Inc., Boulder, CO, USA) using an artificial light source. A LAI-2000 Plant Canopy Analyzer (LAI-2000 PCA; Li-Cor, Lincoln, NE, USA) was used to measure the leaf area index (LAI). Average measurements were taken from 2–3 plots per stand within 0.4 hm² plots. There were 14 stands in the Duke Forest. LAI measurements were acquired in diffuse lighting conditions within an hour after dawn or dusk. A Cimel sun photometer was installed at the NW_Chapel_Hill site (Lat 35°58′15.6″N, Lon 79°5′34.8″W), which was near to the hardwood tower during the campaign. Measurements of aerosol optical depth (AOD) were provided by the level 2.0 quality-assured data of the aerosol robotic network (AERONET), which were obtained from the NW_Chapel_Hill site.

### 2.2.2. Airborne LiDAR Acquisition and Processing

G-LiHT, which is a portable, airborne imaging system with a small footprint scanning LiDAR, a hyperspectral imaging spectrometer and a thermal imager, was used to provide airborne LiDAR acquisition [42]. The acquisition flights of G-LiHT data were carried out above the study site on 30 September in 2013 (Figure 2). Eight flight lines of north–south (N–S) were acquired over the Duke Forest site. To ensure the maps of the duke forest sites were gapless and complete, the swath overlap between flight lines was about 30%. The G-LiHT LiDAR includes a high-performance laser rangefinder with a wavelength at 1,550 nm and rotating polygon three mirror facets with a rotating speed of 100 scans/s. The pulse repetition rates up to 150 kHz in this campaign. The laser beam divergence of LiDAR is 0.3 mrad at the nominal operating altitude of 600 m. Small footprint returns of G-LiHT LiDAR (≤8 pulse⁻¹) could be used to detect small gaps between trees, characterize strip harvesting and provide horizontal distributions of forest stands (Figure 3).

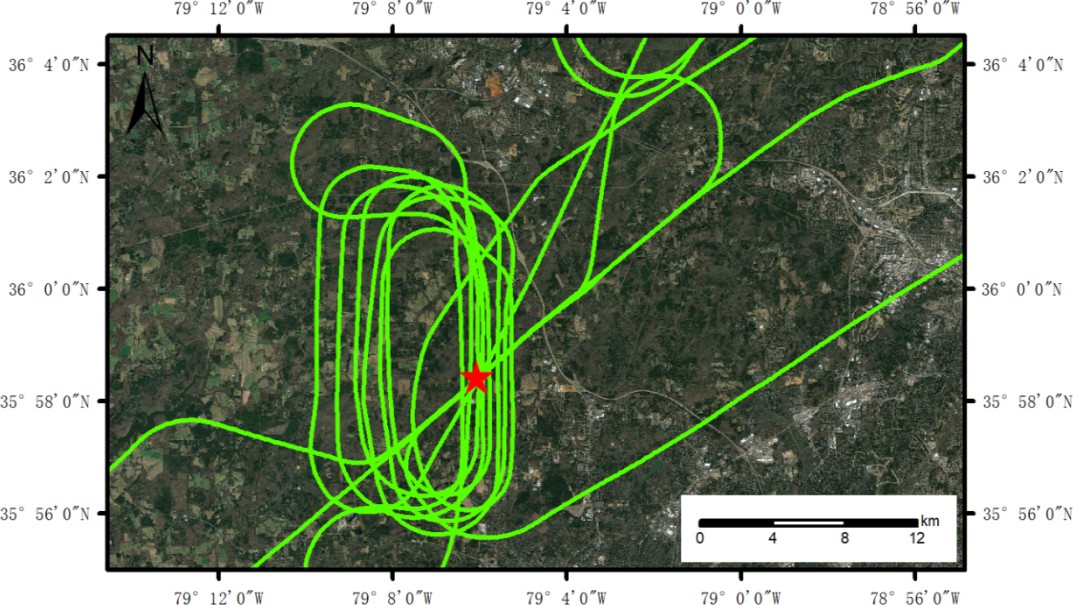

**Figure 2.** The spatial distribution of G-LiHT flight lines acquired over the study site. The red star is the hardwood tower.

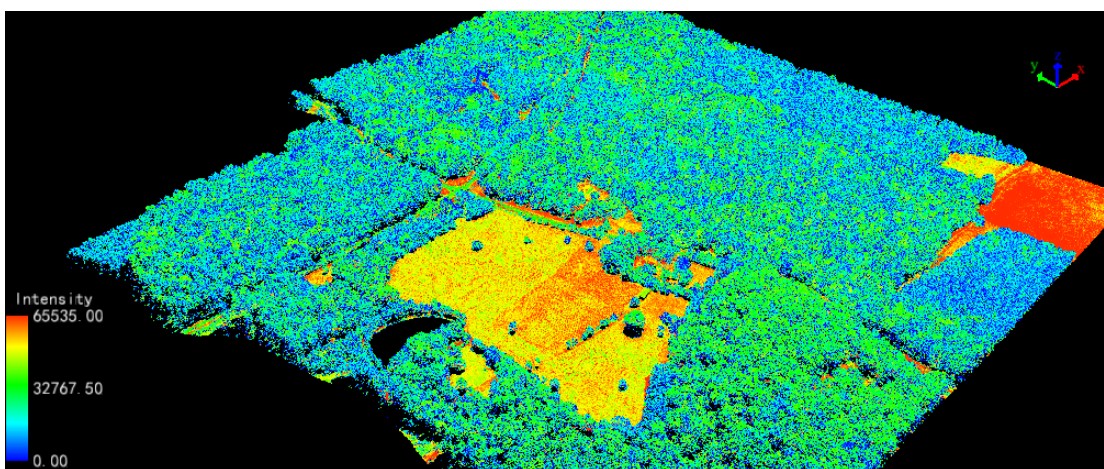

**Figure 3.** The G-LiHT LiDAR returns colored by intensity over the study site on 30 September 2013.

The airborne point clouds were classified into non-ground and ground points. For reconstructing the forest scene geometrically, the airborne laser scanning (ALS) point clouds were segmented using the point cloud segmentation (PCS) algorithm developed by Li et al. [43]. We firstly selected the top of a tree by filtering the highest point in a point set, and then the regional growth from this point was used to judge whether the point clouds below the top of the tree belonged to the same tree. Crown height and height to live crown base then can be derived from the cloud data of each tree. The crown width was calculated with a two-dimensional (2D) convex hull [44]. We first projected the point clouds of a single tree onto a 2D plane. Then, the 2D convex hull of the projected region can be obtained. Finally, the canopy width was calculated from the projected region. The shapes of crown in FluorWPS were predefined as ellipsoids for deciduous trees. Tree crowns were parameterized using crown variables and tree positions.

### 2.2.3. Airborne Hyperspectral Radiance Acquisition and Processing

HyPlant is a hyperspectral imaging system including two optical imaging modules which are mounted in a single platform with a push-broom mode [45]. One of them is a hyperspectral line-imaging spectrometer named dual channel imager (DUAL) with 624 spectral channels covering the spectral ranging from 370 nm to 2500 nm. The FWHM in the VNIR and the SWIR spectral range is 3 nm and 10 nm, respectively. The other is a hyperspectral module named fluorescence imager (FLUO) with 1024 spectral channels covering the spectra ranging from 670 nm to 780 nm. FLUO is designed to measure the vegetation SIF signal and its FWHM is 0.25 nm. Both the field of view (FOV) of these sensors is around 32.3°. The raw radiance images from DUAL and FLUO module were corrected for preprocessing, such as dark current, radiometric calibration and geo-rectification using a dedicated software of CaliGeo [46–48].

The second flight line passed over the study area. Measurements were conducted during cloud-free conditions. The flight height is about 600 m. The swath width is around 380 m. The pixel size of images is 1 m. In this study, we used the image collected between 14:16 and 15:38 (local time) on 30 September 2013 (Figure 4). To evaluate the modeling accuracy of the FluorWPS model, we aggregated radiance images of the DUAL and the FLUO from the original 1 m resolution to 60 m to be representative of canopy scene. The corresponding radiance spectra of the reconstructed canopy scene with LiDAR data were simulated by FluorWPS and evaluated by comparing with the measured spectra by HyPlant for the same area.

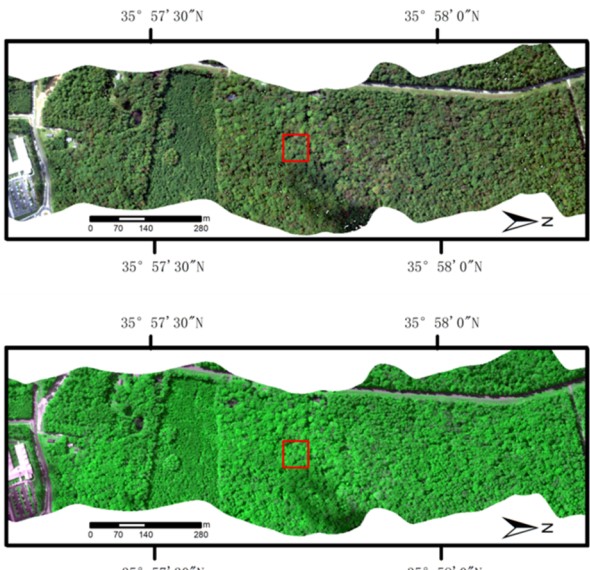

**Figure 4.** Radiance images derived from the DUAL module (top row) and the FLUO module (bottom row) acquired on 30 September 2013. Red rectangle indicates the location of the study area.

### 2.2.4. Datasets Simulated by the DART Model

DART simulates the radiation budget ranging from the visible to the thermal infrared band. By coupling with the Fluspect model, DART can simulate the SIF radiative transfer within homogeneous and complex canopy. DART includes two approaches to describe the scene elements: a 3D juxtaposition of cells with turbid medium and the juxtaposition of facets [49]. For a comparison between the FluorWPS model and the DART model, the optical and structural parameters need to be as consistent as possible. We created a homogeneous scene to compare with the DART model and kept the input parameters of the DART models consistent with that of the FluorWPS model. Furthermore, we created a 3D scene included with turbid trees for further comparison. The inputs required by DART for 1D and 3D canopies are listed in Table 1.

**Table 1.** Inputs required by the DART model for 1D and 3D canopies.

| Parameters | Unit | Value |
|---|---|---|
| **1D scene** | | |
| Cell size x,y,z | m | $1.0 \times 1.0 \times 0.25$ |
| Scene x,y | m | $20.0 \times 20.0$ |
| Leaf area index (LAI) | $m^2 \cdot m^{-2}$ | 3.0 |
| Canopy height | m | 10.0 |
| Leaf angle distribution (LAD) | - | Uniform |
| **3D scene** | | |
| Cell size x,y,z | m | $0.5 \times 0.5 \times 0.5$ |
| Scene x,y | m | $20.0 \times 20.0$ |
| Leaf area index (LAI) | $m^2 \cdot m^{-2}$ | 3.0 |
| Crown height | m | 6.0 |
| Crown diameter | m | 4.0 |
| Crown shape | - | Ellipsoid |
| Leaf angle distribution (LAD) | - | Uniform |
| **Optical properties** | | |
| Carotenoid content ($C_{ca}$) | $\mu g \cdot cm^{-2}$ | 10.0 |
| Equivalent water thickness ($C_w$) | cm | 0.012 |

| | | | |
|---|---|---|---|
| Leaf structure parameter (*N*) | [-] | | 1.8 |
| Fluorescence quantum efficiency (*fqeI*) | [-] | | 0.002 |
| Fluorescence quantum efficiency (*fqeII*) | [-] | | 0.01 |
| **Viewing geometry** | | | |
| Solar zenith angle (SZA) | degree | | 30.0 |
| Solar azimuth angle (SAA) | degree | | 225.0 |
| View zenith angle (VZA) | degree | | 0–70 |
| View azimuth angle (VAA) | degree | | 225 |

### 2.3. Parameterization of the FluorWPS Model

Besides the explicit description of the canopy structure with polygons for relatively short and small plant stands, e.g., crops, the combined geometric primitive with turbid medium method to represent tree crowns, as used in FluorFLIGHT, can also be used in FluorWPS. The crown is approximated as cone, cylinder, ellipsoid or semi-ellipsoid and the trunk as a vertical solid cylinder. The leaves and branches are evenly distributed within the crown. The understory layer is represented by 1D homogeneous media to simulate grass. Similarly, 1D layers with homogeneously distributed atmospheric particles are added above the forest scene to model the atmospheric effect on observed signal received by the sensor at arbitrary height. The parameterization of the FluorWPS model is introduced below.

The FluorWPS model contains five major modules. The main parameters of modules are summarized in Table 2. Canopy module defines a 3D forest canopy scene. Tree crowns are defined by a set of geometric primitives, i.e., cylindrical, conical and ellipsoidal or semi-ellipsoidal objects composed of turbid media, while trunks are approximated as a vertical solid cylinder. Leaves and branches are distributed uniformly within the crown. The understory layer of the scene is described by 1D homogeneous media such as grass. Viewing geometry module offers an ideal sensor, which is located at a predefined height and direction, to collect fluorescence and non-fluorescence radiance from the canopy. The atmosphere module tracks the path of photons through the atmosphere, enabling Fluor-WPS to simulate radiance at any height. It uses the extinction coefficient, single scattering albedo and scattering phase function of atmospheric molecules and aerosols to describe the atmospheric condition. Light source module includes isotropic skylight and direct sunlight. The direction of the direct sunlight is determined by the solar azimuth angle (SAA) and the solar zenith angle (SZA). Optical properties module aims to parameterize spectral properties of elements within the scene. These spectral properties include transmittance and reflectance as well as excitation-fluorescence matrices (EF-matrices). Figure 5 presented the scheme of FluorWPS parameterization. The detailed parameterization processes are described as follows.

**Table 2.** Inputs required by the FluorWPS model.

| Module | Parameter | Unit | Source (in This Work) |
|---|---|---|---|
| Canopy | Geometry coordinates | m | Airborne Laser Scanning |
| | Geometry radius | m | Airborne Laser Scanning |
| | Leaf area index (LAI) | $m^2 \cdot m^{-2}$ | Field measurements |
| Viewing geometry | Sensor altitude | m | Sensor overpass |
| | View zenith angle (VZA) | degree | Nadir observation |
| | View azimuth angle (VAA) | degree | Sensor overpass |
| Atmosphere | Atmospheric extinction coefficient | $m^{-1}$ | Simulated by MODTRAN |
| | Single scattering albedo | % | Simulated by MODTRAN |
| | Scattering phase function | [-] | Simulated by MODTRAN |
| Light source | Solar irradiance | $mW \cdot m^{-2} \cdot nm^{-1}$ | Simulated by MODTRAN and SCOPE |

| | Solar zenith angle (SZA) | degree | Solar calculator |
|---|---|---|---|
| | Solar azimuth angle (SAA) | degree | Solar calculator |
| Optical properties | Tree leaf reflectance | % | Field measurements |
| | Tree leaf transmittance | % | Field measurements |
| | Grass leaf reflectance | % | Assumed equal to tree leaf reflectance |
| | Grass leaf transmittance | % | Assumed equal to tree leaf transmittance |
| | Soil reflectance | % | ENVI spectral library |
| | Bark reflectance | % | Assumed equal to half of tree leaf reflectance |
| Tree leaf backward and forward EF-matrices for Photosystem I | | [-] | Simulated by Fluspect |
| Tree leaf backward and forward EF-matrices for Photosystem II | | [-] | Simulated by Fluspect |
| Grass leaf backward and forward EF-matrices for Photosystem I | | [-] | Assumed equal to tree leaf for Photosystem I |
| Grass leaf backward and forward EF-matrices for Photosystem II | | [-] | Assumed equal to tree leaf for Photosystem II |

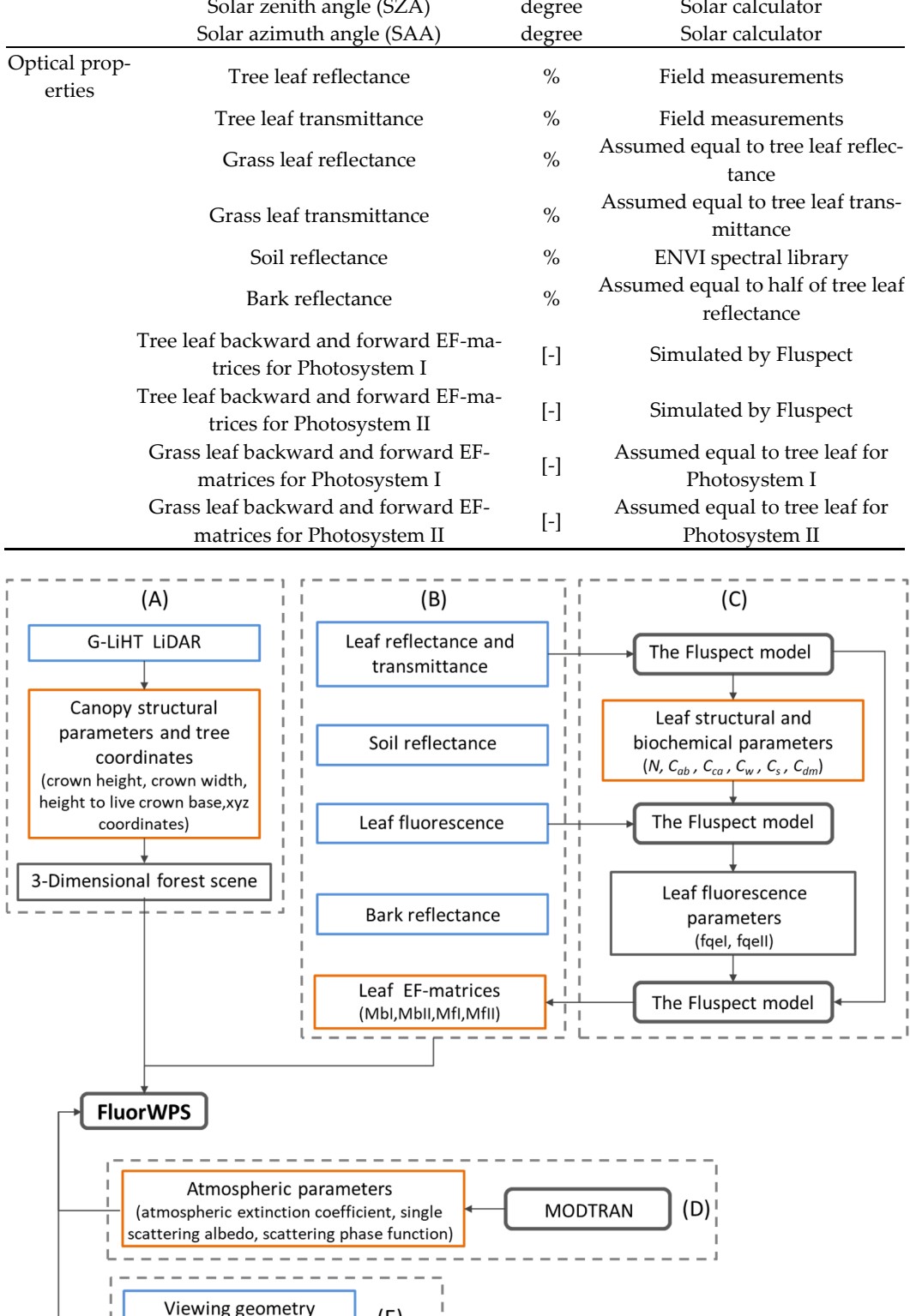

**Figure 5.** The scheme of FluorWPS parameterization includes six parts: (**A**) canopy module, (**B**) optical properties module, (**C**) model inversion process for excitation-fluorescence matrices, (**D**) atmosphere module and (**E**) viewing geometry module. Measured parameters were put in the blue boxes, and computed parameters were put in the yellow boxes.

The Forest scene was reconstructed according to canopy parameters (e.g., crown height, crown width, height to live crown base) obtained from G-LiHT point clouds. The

scene was made up of deciduous trees and an understory layer of grass. The size of the reconstructed scene was 60 m × 60 m × 40 m. The crowns of reconstructed trees in Fluor-WPS were predefined as ellipsoids filled by turbid media. The leaf angle distribution (LAD) of tree leaves was assumed to be planophile, which was more suitable for describing the LAD of deciduous trees [50]. The grass layer (LAI assumed to 1.0) was created with a spherical leaf angle distribution.

Tree leaves and grass leaves were parameterized using optical properties obtained in Section 2.2.1. Due to the predominant canopies of the study site being hickory, we computed the average value as the inputs to considering the spatial representativeness of the leaves. Grass leaves were given the same fluorescence and optical parameters as the tree leaves for simplicity. Trunk reflectance was empirically set as half of the tree leaf reflectance. The soil in the Duke Forest is rich in the minerals. Therefore, the corresponding soil reflectance was selected from the ENVI 5.3 spectral library. Spectral data were shown in Figure 6a. We used the Fluspect model to retrieve the four EF-matrices [32]. The module C in Figure 5 described the two sections of inversion method. At the first section, tree leaf reflectance as well as transmittance were used to retrieve six biochemical parameters, i.e., chlorophyll content ($C_{ab}$), senescent material ($C_s$), dry matter content ($C_{dm}$), carotenoid content ($C_{ca}$), water content ($C_w$) and leaf structure parameter ($N$), by inverting Equation (1). The inverted leaf optical parameters were listed in Table 3.

$$Fp(N, C_{ab}, C_{ca}, C_w, C_s, C_{dm}) = \sum_{\lambda \in [400, 2, 500]} \left\{ \begin{array}{l} [\rho_l(\lambda) - \rho_{simu}(N, C_{ab}, C_{ca}, C_w, C_s, C_{dm}, \lambda)]^2 \\ + [\tau_l(\lambda) - \tau_{simu}(N, C_{ab}, C_{ca}, C_w, C_s, C_{dm}, \lambda)]^2 \end{array} \right\} \quad (1)$$

where $\tau_l$ and $\rho_l$ are the measured tree leaf transmittance and reflectance, respectively, and $\tau_{simu}$ and $\rho_{simu}$ are the simulated tree leaf transmittance and reflectance, respectively.

**Table 3.** The leaf optical parameters retrieved from the Fluspect model.

| Parameters | Unit | Value |
|---|---|---|
| Chlorophyll content ($C_{ab}$) | μg·cm$^{-2}$ | 31.4 |
| Senescent material ($C_s$) | [-] | 0.187 |
| Dry matter content ($C_{dm}$) | g·cm$^{-2}$ | 0.0017 |
| Carotenoid content ($C_{ca}$) | μg·cm$^{-2}$ | 10.7 |
| Water content ($C_w$) | g·cm$^{-2}$ | 0.006 |
| Leaf structure parameter ($N$) | [-] | 1.79 |
| Fluorescence quantum efficiency (*fqeI*) | [-] | 0.0016 |
| Fluorescence quantum efficiency (*fqeII*) | [-] | 0.019 |

In the second section, we fixed the six biochemical parameters in Table 3 to invert fluorescence quantum efficiency of PSI (*fqeI*) and PSII (*fqeII*). Then, the four EF matrices were retrieved from the Fluspect model.

The direct and diffuse irradiance were simulated by MODTRAN. We chose the rural aerosol model with a visibility of 23 km and the mid-latitude summer gas model as standard inputs to obtain the MODTRAN5.tp7 file using the MODTRAN Interrogation Technique (MIT) [51]. The examples of the simulated spectra are shown in Figure 6b.

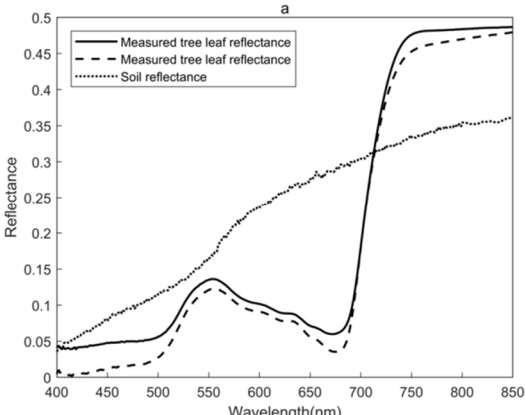 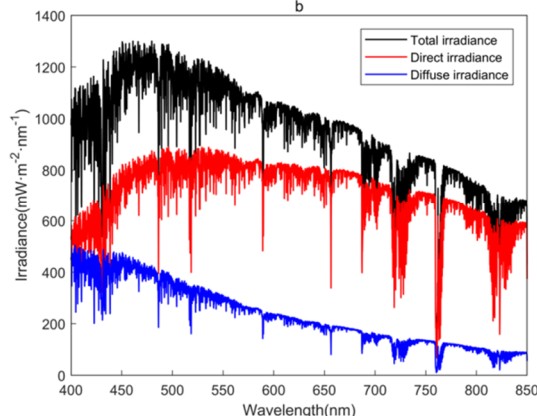

**Figure 6.** Examples of spectra for parameterization: (**a**) the reflectance spectra of soil and the measured reflectance and transmittance of tree leaves; (**b**) the simulated total, direct and diffuse irradiance.

We used MODTRAN to simulate the atmospheric conditions when the HyPlant flew over the study area. According to the measurement of the NW_Chapel_Hill site at seven wavelengths (340, 380, 440, 500, 675, 870 and 1020 nm), we calculated the aerosol optical depth at 550 nm. Then, MODTRAN calculated atmospheric parameters using an aerosol optical depth at 550 nm. Sensor height was set according to the flight height of the HyPlant. Sun-target-viewing geometry was calculated used solar calculator based on the date and time of the HyPlant acquisition.

### 2.4. Model Local Sensitivity Analysis

The FluorWPS model describes the process of photons emitted from the light source and spread to the sensor. Foliage area volume density (FAVD) determines whether photons collide with elements (leaves, branches or grass) or not. LAD determines the direction of photons emission after colliding with elements. Sun-target-viewing geometry has an important impact on the generation and collection of energy. Fractional vegetation cover (FVC) describes the heterogeneity of the forest scene, which has an implication for the retrieval and interpretation of SIF. The LAI of the understory influences the magnitude of SIF after photons collide with the understory.

We identified the relative importance of the canopy structural parameters to the red and far-red SIF using the local sensitivity analysis at the solar principal plane. Table 4 listed the canopy structural parameters of the scene. We also investigated the influence of SZA variations on SIF and analyzed the multi-angular SIF distributions. For characterizing the SIF response to changing SZA, we used the anisotropy index (*ANIX*) in Equation (2) to evaluate the amplitude of SIF variations [52]. The *ANIX* was originally defined as ratio of the maximum to the minimum reflectance values of the solar principal plane [53]. The range of the simulated spectra increased from 640 nm to 850 nm and the increment is 1 nm. View azimuth angle (VAA) was set from 0° to 350° with increments of 10°. View zenith angle (VZA) was set from 0° to 70° with increments of 5°. The number of the simulated viewing directions was 505. SZA were fixed at 30° and SAA were set at 140° for the subsequent sensitivity analysis. The negative value of VZA corresponded to the forward viewing direction and the positive value of VZA corresponded to the backward viewing direction. In the FluorWPS model, the sensor receives the total SIF, which includes the emission part that comes from leaves and the scattering part that comes from leaves and soil. We mainly investigated the sensitivity analysis of the total SIF. In some cases, we also analyzed total SIF in two parts:

$$ANIX = \frac{SIF_{max}}{SIF_{min}} \tag{2}$$

where $SIF_{max}$ and $SIF_{min}$ are the maximum and minimum value of the measured SIF in the solar principal plane, respectively.

**Table 4.** Canopy structural parameters of FluorWPS used in the sensitivity analysis.

| Parameters | Unit | Stand Value | Rang of Variation |
|---|---|---|---|
| Foliage area volume density (FAVD) | m⁻¹ | 3.5 | 0.5–4.5 |
| Leaf angle distribution (LAD) | degree | planophile | uniform/extremophile/ plagiophile/spherical/ erectophile/planophile |
| Fractional vegetation cover (FVC) | % | 76 | 39–95% |
| Leaf area index of understory (LAI) | m²·m⁻² | 1 | 0–2 |

### 3. Results

#### 3.1. Reconstruction of the 3D Forest Scene

The result of the G-LiHT LiDAR reconstruction was represented as a 3D forest scene, which was made up of 90 trees. The LAI and the individual trees of the 60 × 60 m scene are shown in Figure 7. The qualitative comparison between the reconstructed scene (Figure 7a) and the radiance image (Figure 7c) showed that the spatial distribution of the canopy gaps was similar, but the gaps in the reconstructed scene were more obvious. The LAI value of hardwood tower in situ measurements is 4.6 m² m⁻². We used this value as a reference for validating the accuracy of the reconstructed forest scene. The mean LAI value of the study area derived from G-LiHT LiDAR was 4.76 m² m⁻² (Figure 7d). There was a relative deviation of 3.5%, which denoted that the structural characteristics of the reconstructed forest scene is in accordance with that of the realistic forest scene. The reconstructed scene shows good consistency with the realistic forest scene using a cross-comparison HyPlant image and LAI measurement.

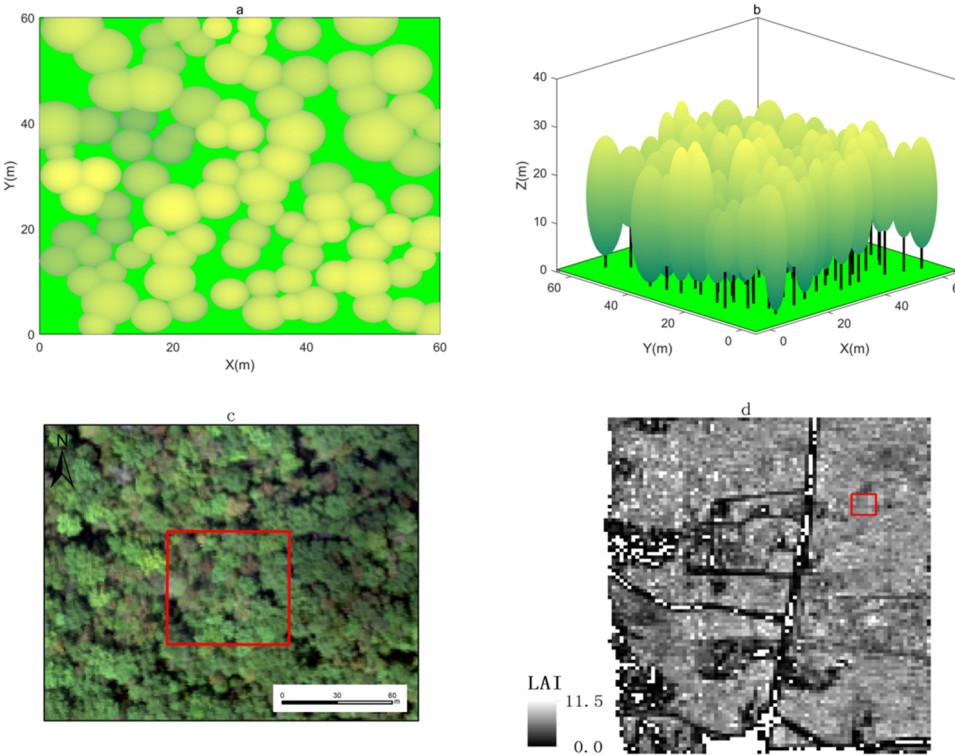

**Figure 7.** The reconstructed forest scene using G-LiHT LiDAR: (**a**) the nadir view and (**b**) the side view. The images of the forest scene: (**c**) the radiance image from HyPlant DUAL and (**d**) LAI image derived from G-LiHT LiDAR. Rectangles indicate the location of the study area.

### 3.2. Comparison with HyPlant Measurements

To evaluate the performance of the FluorWPS simulation for a representative forest scene, we aggregated the radiance spectra for the selected study area with 60 × 60 pixels. Then, we compared the radiance spectra simulated by FluorWPS with the corresponding aggregated radiance spectra of HyPlant (Figure 8). The general shapes as well as values of simulated radiance spectra agreed fairly well with those measured by HyPlant, though there were some differences. For the DUAL sensor, the comparisons showed that the root mean square error (RMSE) and the coefficient of determination ($R^2$) between the measured and simulated radiance spectra were 1.85 mW·m$^{-2}$·sr$^{-1}$·nm$^{-1}$ and 0.998, in the range of 400–800 nm with steps of 3 nm. Conversely, for the FLUO sensor, the RMSE and $R^2$ between the simulated and measured radiance spectra were 9.12 mW·m$^{-2}$·sr$^{-1}$·nm$^{-1}$ and 0926, in the range of 670–782.35 nm with steps of 0.25 nm. Underestimation in the range of 400–502 nm, and 772–800 nm for the DUAL, was found, with mean relative differences of 21.8% and 3.2%, respectively. Overestimation in the range of 742–759 nm for the FLUO was found, with a mean relative difference of 5.3%. Generally, these results suggest that the modeling accuracy is reasonable.

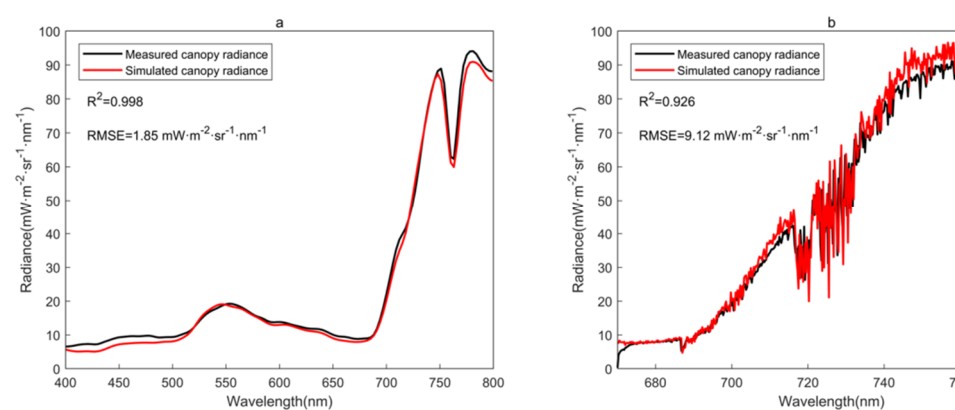

**Figure 8.** Comparison of the radiance measured by HyPlant with those simulated by FluorWPS: (**a**) for the DUAL sensor and (**b**) for the FLUO sensor.

### 3.3. Comparison with the DART Model

We compared both the nadir and multi-angular TOC SIF at the principal plane by the FluorWPS model and the DART model for 1D and 3D canopies. In the nadir direction, the SIF of the 1D canopy simulated by the FluorWPS model agreed well with the DART model, with an $R^2$ of 0.999 and an RMSE of 0.024 mW·m$^{-2}$·sr$^{-1}$·nm$^{-1}$, respectively (Figure 9a). At different view directions, the SIF of the 1D canopy simulated by the FluorWPS model at both 685 nm and 740 nm showed an agreement with the values of the DART model, with an $R^2$ of 0.907 and 0.948, and an RMSE of 0.060 mW·m$^{-2}$·sr$^{-1}$·nm$^{-1}$ and 0.122 mW·m$^{-2}$·sr$^{-1}$·nm$^{-1}$, respectively (Figure 9b). In the nadir direction, the SIF of the 3D canopy simulated by the FluorWPS model was slightly overestimated at 740 nm, with an $R^2$ of 0.995 and an RMSE of 0.071 mW·m$^{-2}$·sr$^{-1}$·nm$^{-1}$, respectively (Figure 9c). At different view directions, the SIF of the 3D canopy simulated by the FluorWPS model at both 685 nm and 740 nm was generally consistent with the values of the DART model, with an $R^2$ of 0.901 and 0.969 and an RMSE of 0.070 mW·m$^{-2}$·sr$^{-1}$·nm$^{-1}$ and 0.071 mW·m$^{-2}$·sr$^{-1}$·nm$^{-1}$, respectively (Figure 9d). The SIF of the FluorWPS model near the hotspot was overestimated for the 3D canopy. The comparison shows that there was a high consistency of TOC SIF between the FluorWPS model and the DART model for 1D and 3D canopies.

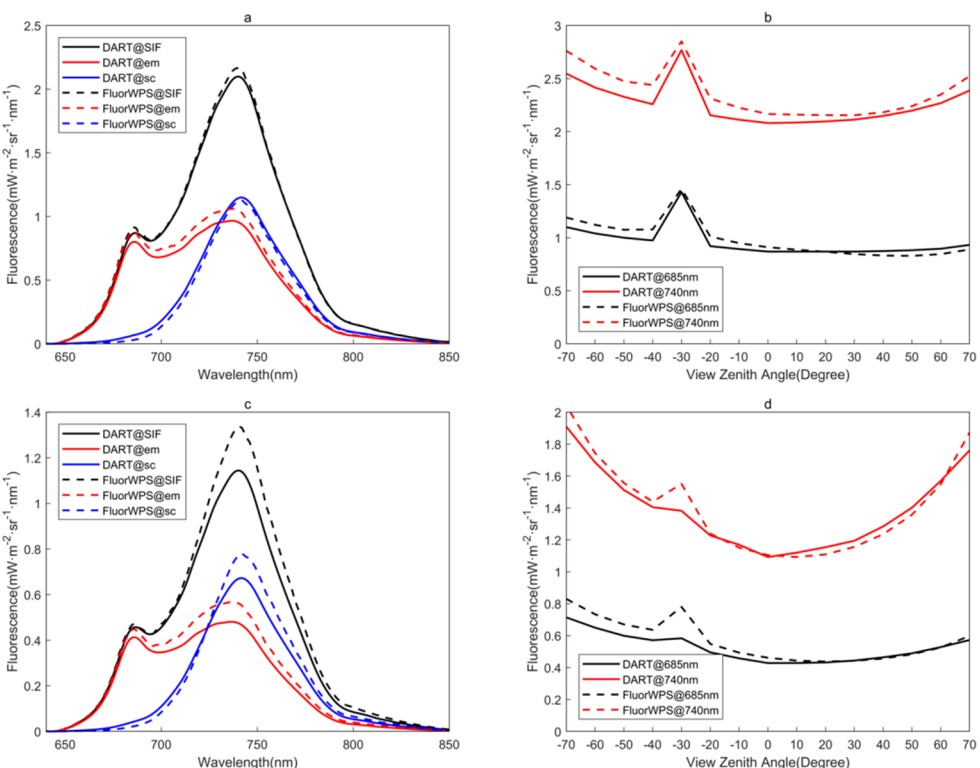

**Figure 9.** Comparison of the SIF measured by FluorWPS with those simulated by DART: (**a**) for the nadir view of 1D canopy, (**b**) for the different view directions of 1D canopy, (**c**) for the nadir view of 3D canopy and (**d**) for the different view directions of 3D canopy.

*3.4. Local Sensitivity Analysis of TOC SIF*

3.4.1. Impact of the Foliage Area Volume Density on TOC SIF

We set the FAVD value from 0.5 to 4.5 with steps of 1 to investigate the effect of FAVD on TOC SIF. In the nadir direction, the spectra of SIF increased at first rapidly with FAVD (from 0.5 to 2.5), which then increased slowly (from 2.5 to 4.5) at different wavelengths. When the FAVD changed from 0.5 to 4.5 at 685 nm and 740 nm, the SIF increased by 45% and 47%, respectively (Figure 10a). At different view directions, SIF near the hotspot became saturated with FAVD increased and the SIF gradually converged when the VZA was large (Figure 10b,c). The results indicate that FAVD plays an obvious effect on the magnitude of TOC SIF.

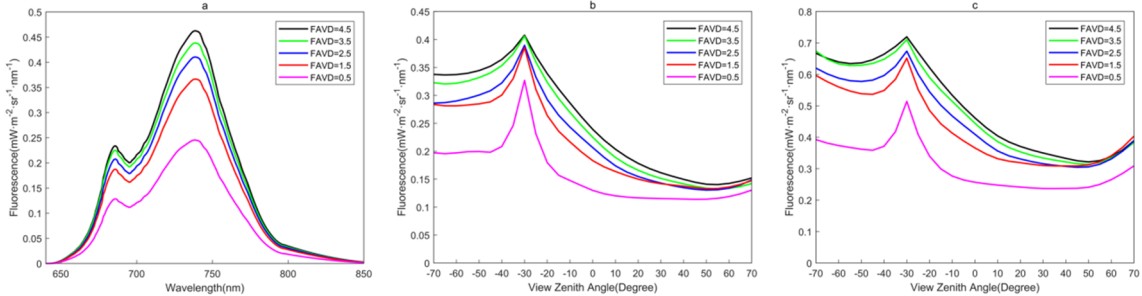

**Figure 10.** Total SIF spectra for different FAVD values simulated by the FluorWPS model: (**a**) SIF with a nadir view, (**b**) SIF at 685 nm with different view directions and (**c**) SIF at 740 nm with different view directions.

### 3.4.2. Impact of the Leaf Angle Distribution on TOC SIF

We evaluated six representative LAD values to analyze the impact of LAD on TOC SIF. In the nadir direction, the erectophile LAD had the largest SIF at different wavelengths (Figure 11). The emission part of the planophile LAD had the largest contribution to the total SIF reaching 72%, while the erectophile LAD had the smallest contribution for only 48% at 740 nm (Figure 11b). The planophile LAD and the erectophile LAD had a reverse effect for the scattering part at 740 nm (Figure 11c). At different view directions, the erectophile LAD exhibited a significant bowl-like shape, while the planophile LAD was not obvious (Figure 11d,e). The observations indicate that LAD has a considerable impact on TOC SIF as a result of different scattering ability and escape probability.

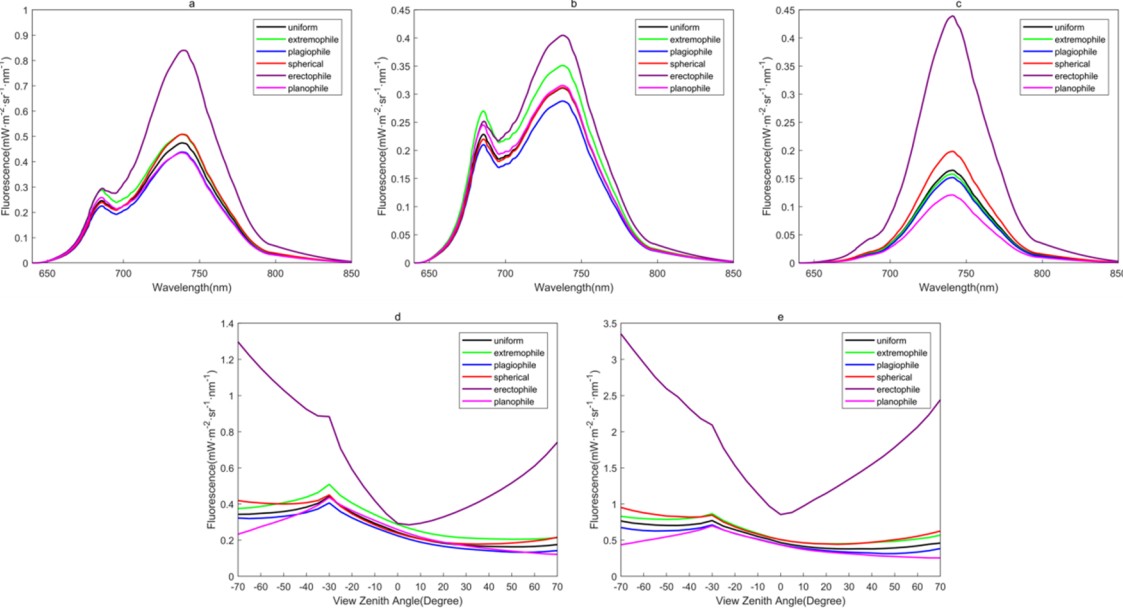

**Figure 11.** SIF spectra for different LAD values simulated by the FluorWPS model: (**a**) total SIF with a nadir view, (**b**) the emission part of SIF with a nadir view, (**c**) the scattering part of SIF with a nadir view, (**d**) SIF at 685 nm with different view directions and (**e**) SIF at 740 nm with different view directions.

### 3.4.3. Impact of the Fractional Vegetation Cover on TOC SIF

We selectively removed trees from the scene to evaluate the effect of FVC on TOC SIF. In the nadir direction, SIF at 740 nm increased more markedly than that at 685 nm with an increase of the FVC value. When the FVC value increased from the minimum to the maximum, the SIF differed by 48% and 46% at 685 nm and 740 nm, respectively. At different view directions, SIF near the hotspot converged and the bowl-like shape of TOC SIF became evident with the increase of FVC (Figure 12b,c). Our results indicate that FVC has a considerable effect on the magnitude and the shape response of TOC SIF.

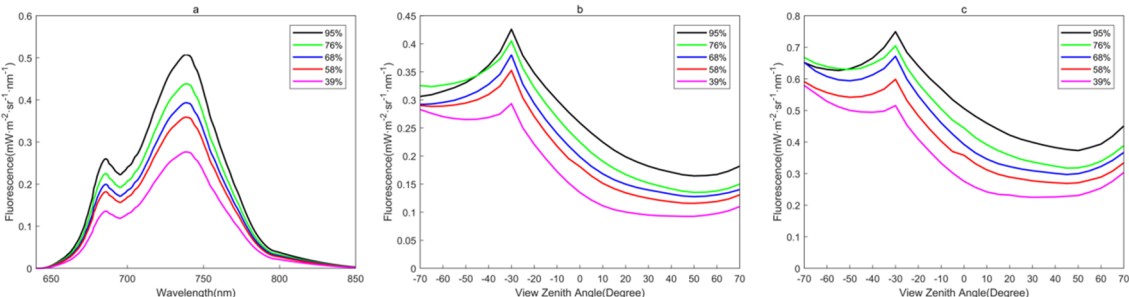

**Figure 12.** Total SIF spectra for different FVC values simulated by the FluorWPS model: (**a**) SIF with a nadir view, (**b**) SIF at 685 nm with different view directions and (**c**) SIF at 740 nm with different view directions.

### 3.4.4. Impact of the Understory on TOC SIF

We investigated the changes of understory LAI on TOC SIF. We set LAI of the understory equal to zero as a reference then changed the LAI value from 0.5 to 2 with steps of 0.5. In the nadir direction, SIF increased with 5.2% and 7.6% at 685 nm and 740 nm for an LAI equal to 0.5, and 11.8% and 12.9% for an LAI equal to 2, respectively (Figure 13a). SIF tended to be converged with an increased LAI value, and the change of SIF at 740 nm is more significant than that at 685 nm. At different view directions, the understory vegetation mainly affected the SIF near the hotspot. The understory with no SIF emission had a more remarkable bowl-like shape (Figure 13b,c). The results demonstrate that the understory had a non-negligible influence on TOC SIF at different wavelengths and at different view directions.

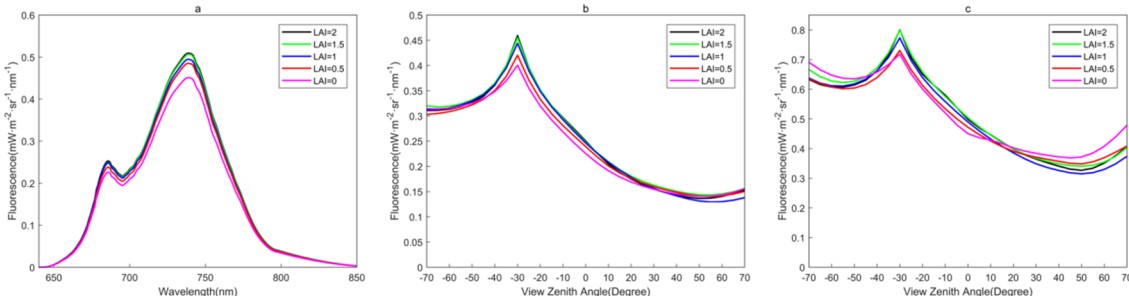

**Figure 13.** Total SIF spectra for different understory LAI values simulated by the FluorWPS model: (**a**) SIF with a nadir view, (**b**) SIF at 685 nm with different view directions and (**c**) SIF at 740 nm with different view directions.

### 3.4.5. Impact of Solar Zenith Angle on TOC SIF

We changed the value of SZA from 0° to 70° with steps of 10° for investigating the influence of the illuminance on SIF. When the value of SZA increased, SIF in the forward direction increased significantly, while it decreased in the backward direction. Moreover, the position of the hotspot moved to a forward viewing direction coinciding with SZA (Figure 14). $F_{740}\_ANIX$ and $F_{685}\_ANIX$ increased with SZA, while $F_{685}\_ANIX$ was greater than $F_{740}\_ANIX$ with the same SZA. When the SZA was 70°, ANIX reached its maximum value (Table 5). This result indicated that SZA was an important factor affecting the shape of SIF directional response.

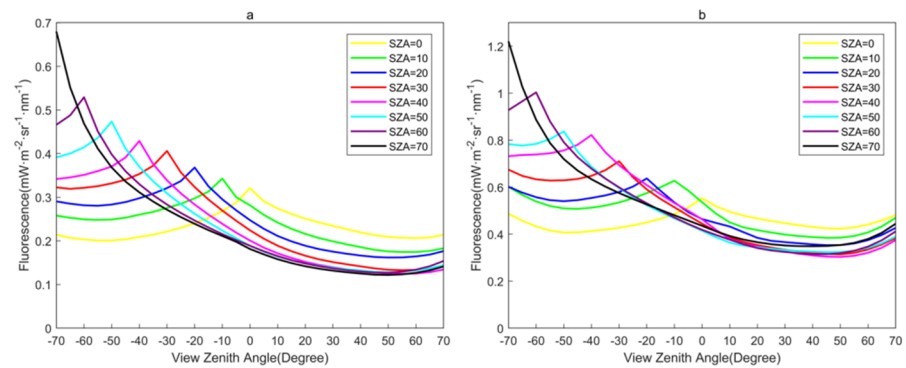

**Figure 14.** Total SIF spectra for 685 nm (**a**) and 740 nm (**b**) simulated by the FluorWPS model at different SZA values.

**Table 5.** ANIX calculated for $F_{740}$ and $F_{685}$ with different SZA.

| SZA (°) | 0 | 10 | 20 | 30 | 40 | 50 | 60 | 70 |
|---|---|---|---|---|---|---|---|---|
| $F_{740}\_ANIX$ | 1.36 | 1.63 | 1.80 | 2.26 | 2.71 | 2.58 | 3.18 | 3.50 |
| $F_{685}\_ANIX$ | 1.60 | 1.98 | 2.30 | 3.04 | 3.61 | 3.69 | 4.65 | 5.57 |

### 3.4.6. Polar Maps of TOC SIF at the Two Peaks

Figure 15 presented the polar maps of red and far-red SIF simulated by FluorWPS. The nadir viewing direction (VZA equal to 0°) is in the center of Figure 15. VZA increased outward from 0° to 70° and VAA increased clockwise from 0° to 350°. The distribution of the total SIF displayed a near symmetrical distribution at the principal plane. Both the red and far-red SIF simulated by the FluorWPS model showed a bowl-like shape, and there was a local maximum value coinciding with the hotspot direction. The plots indicate that the red and far-red SIF, at different view directions, are significantly affected by the canopy structure.

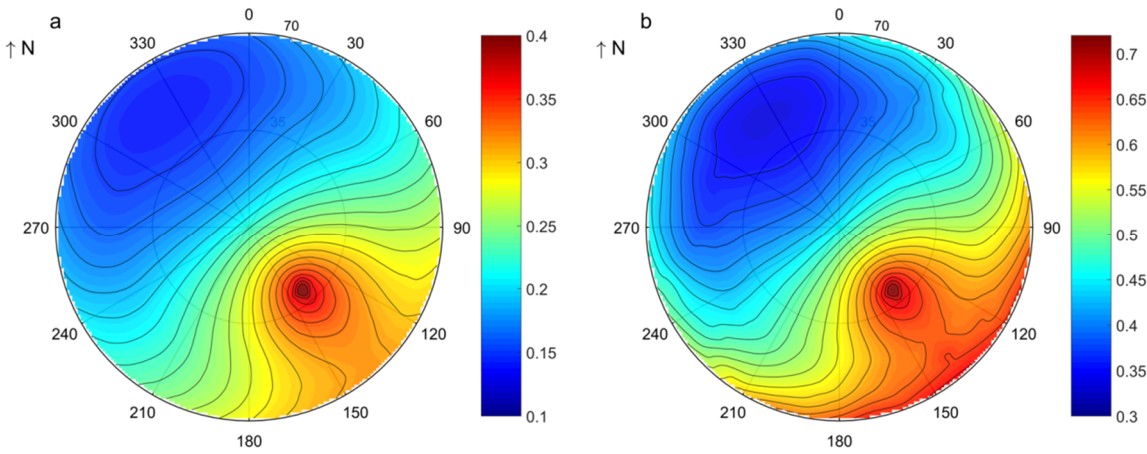

**Figure 15.** Polar maps of the total SIF simulated by FluorWPS for 685 nm (**a**) and 740 nm (**b**).

### 4. Discussion

We used ALS data to reconstruct a forest scene and tested the reliability of the reconstructed forest scene qualitatively and quantificationally. A qualitative comparison between the reconstructed forest scene and the radiance image of DUAL showed that using ellipsoidal shapes to describe crowns generally results in a gap fraction that is obvious (Figure 7). This is attributed to the clustered effect of plant material in the mostly closed canopy layer. The quantitative comparisons between the retrieved LAI and the measured validated that the reconstructed scene coincided with the real stand. Due to a lack of accurate measurements of soil spectra and stand parameters such as understory spectra and LAD, the uncertainties lead to the relative deviation. Comparisons between the simulated radiance spectra and the measured validated the capability of the FluorWPS model to simulate radiance. The simulated radiance spectra of FluorWPS showed vegetation radiance characteristics and atmospheric absorption features as expected (Figure 8). We observed some differences between the simulated radiance spectra and the measured spectra. There are four possible reasons for these differences. Firstly, FluorWPS does not take the leaf traits into account, but leaf optical properties vary considerably among different types and stand ages. FluorWPS describes crowns as simple ellipsoidal shapes filled with turbid media, which are different from realistic canopies. Secondly, there are a substantial number of shrubs and clustered trees in a complex forest ecosystem, which may be difficult to detect. Thirdly, there are instrumental measurement errors for obtaining leaf spectral data and LiDAR data. The inversion algorithms for EF-matrices and canopy structural parameters may also introduce errors. Finally, FluorWPS does not consider the instrumental characteristics of HyPlant, such as the signal-to-noise ratio (SNR) and modulation-transfer function (MTF), which may also lead to bias. A comparison with the DART model enhanced the reliability of the FluorWPS model (Figure 9). There were some discrepancies between the two models, especially for the 3D canopy. These differences were attributed

to the different parameterized methods for canopy scene and mathematical approaches for modeling the radiative transfer from the leaf to the canopy.

Sensitivity analyses by the FluorWPS model suggested that canopy structural parameters could drive substantial changes on SIF. The sensitivity analysis of FAVD demonstrates that it is an important aspect of canopy structural parameters. When the FAVD value ranges from 0.5 to 2.5, photons mainly collide with branches within the canopy. With an FAVD value between 2.5 and 4.5, photons collide with leaves to excite fluorescence. The increase of the FAVD values may enhance the ability of re-absorption of the red SIF within the canopy; thus, the increase of the red SIF was slightly lower than that of the far-red SIF (Figure 10). Because the directional escape ratios are affected by the canopy structure, so the magnitude and shape of SIF at different wavelengths and view directions varied with the changes of LAD value (Figure 11). Flat leaves will produce a higher leaf efficiency of light interception. Thus, the emission ratio of SIF for the planophile LAD is higher than that for the erectophile LAD. There was a noticeable deviation in the NIR regions for erectophile LAD. This mainly comes from multiple scattering because emission part agrees closely with each other. Our results support the published studies about the importance of considering horizontally and vertically heterogeneous variations of LAD when we interpret SIF [54,55]. Forest ecosystem is a typical complex system, the most widely distributed forms of which are tree-grass and tree-shrub ecosystem [56]. The sensitivity analysis of FVC suggests that heterogeneity of scene can drive substantial differences on SIF (Figure 12). The FluorWPS model is suitable for SIF modeling with significant heterogeneity scene. Understory layer is an important factor even for dense canopy scene (FAVD = 3.5 m$^{-1}$, FVC = 76%) (Figure 13). The sensitivity analysis result is consistent with the conclusion obtained in previous studies that understory vegetation has a notable influence on TOC reflectance and SIF signal [57–59]. At different view directions, SIF emitted from the discontinuous 3D canopies is significantly affected by FVC and understory LAI. Due to different properties of grass and tree, changes in the proportions of trees and grass in the FOV will affect SIF at different view angles. This truly reflects the optical response of the ecosystem.

The magnitude and shape of SIF were affected in varying illumination conditions (Figure 14). The magnitude of SIF was higher in the forward direction than in the backward direction. Because the changes of the ratio of shaded and sunlit leaves in the FOV are caused by the changing SZA [60,61]. The red SIF had a similar shape to the far-red SIF but with higher variability (higher *ANIX*) (Table 5), which partly attributed to the re-absorption in the red region [62,63]. The increase of the far-red SIF at high VZAs is caused by photons scattering (Figure 15). The directional response of SIF is caused by the canopy structure, which controlled the absorption and scattering of the light within the canopy [64]. The effects of the varying viewing geometries can be explained by the changes of the path length of the fluorescence photons that escaped from the canopy to the direction of the sensor. The escape path length of SIF photons becomes longer when VZAs are high. The far-red SIF photons are likely to have multiple scattering within the canopy.

In summary, our work described the results of the FluorWPS model evaluation conducted at the Duke Forest study site and tested the sensitivity of TOC SIF to canopy structural parameters and sun-target-viewing geometry. Although there are many factors influencing the simulations, the comparisons between the simulated and measured radiance spectra agreed well. Due to the complexity for the model evaluation, the comparison of SIF between FluorWPS and DART was still satisfactory in spite of some discrepancies. The SIF comparison between the spectra retrieved from the image against the simulated spectrais a research topic for a future study. The sensitivity analysis disentangled that canopy structural parameters and sun-target-viewing geometry had an evident impact on TOC SIF. Our study assumed that trees had constant spectral and canopy structural parameters throughout the scene. It has been reported that leaf structure and physiology were heterogeneous between different forest stands [65]. This can parameterize the key leaf spectral and canopy structural parameters for different types of forest stands in future

research. The integrated modeling method of FluorWPS and G-LiHT LiDAR developed in this work could improve the interpretation of SIF variations related to the viewing geometry and canopy structural factors from ground, airborne and satellite observations.

## 5. Conclusions

We evaluated the capability of the FluorWPS model to simulate at-sensor hyperspectral radiance received by an airborne sensor as well as SIF at the TOC level. Firstly, we conducted a modeling study based on G-LiHT LiDAR data to reconstruct the forest scene. The relative error was 3.5% between the retrieved and the measured LAI of the reconstructed scene. Secondly, we evaluated the modeling accuracy by comparing the simulated radiance spectra with the measured ones by the two high-performance imaging spectrometers of HyPlant for the same area. The performance of FluorWPS showed that the simulated radiance spectra represented well ranges and variations of radiance for the DUAL ($R^2 = 0.998$) and the FLUO ($R^2 = 0.926$). In particular, the radiance spectra simulated by FluorWPS showed good atmospheric absorption characteristics. Thirdly, we compared TOC SIF simulated by the FluorWPS model with that of the DART model. There was a close agreement between the two 3D radiative transfer models. For the 1D canopy, the $R^2$ of the nadir observed an SIF of 0.999, and an $R^2$ of the multi-angular SIF of 0.907 at 685 nm and 0.948 at 740 nm, respectively. For the 3D canopy, the $R^2$ of the nadir observed SIF of 0.995, and the $R^2$ of the multi-angular SIF of 0.901 at 685 nm and 0.969 at 740 nm, respectively. We conclude that the FluorWPS model has a potential for simulating the SIF for complex structural canopy.

We also analyzed the impacts of the canopy structural parameters and sun-target-viewing geometry on the magnitude and directional response of TOC SIF in a deciduous forest. Firstly, the maximum difference of the observed SIF can be as large as 45% and 47% at 685 nm and 740 nm for different FAVDs, and 48% and 46% for FVCs, respectively. LAD has a markedly influence on the magnitude of SIF, with a ratio of emission to the total SIF range from 0.48 to 0.72. The understory is also an important factor that influenced TOC SIF, even for a dense forest canopy (FAVD = 3.5 $m^{-1}$, FVC = 76%), with a contribution of 10%+ more. Secondly, the red SIF had a similar shape to the far-red SIF but with higher variability (higher *ANIX*) in varying illumination conditions. The sensitivity analyses revealed that the canopy structural factors and sun-target-viewing geometry had significant effects on the heterogeneity both in horizontal and vertical dimensions of forest canopies. Future improvements of the FluorWPS model include a modeling and parameterization method for different type forest stands.

The FluorWPS model provides a feasible way to simulate radiance spectra considering the atmospheric contribution. The integrated modeling method of the FluorWPS model and G-LiHT LiDAR is able to simulate most spectral features of hyperspectral radiance received by an airborne sensor properly. This is the first systematical evaluation of 3D RTMs using airborne hyperspectral remote sensing measurements and concurrent in situ measurements. Our study paves the way for the interpretation of SIF at different scales and angular configurations in a terrestrial ecosystem.

**Author Contributions:** Original draft preparation and formal analysis, C.T.; review and editing, Y.B. and F.Z.; software, F.Z., C.F. and Z.L.; validation, Q.H. All authors have read and agreed to the published version of the manuscript.

**Funding:** This research was supported by the National Science Foundation of China (Grant Nos. 41401410, 41611530544 and 41771382).

**Acknowledgments:** The authors thank the FLEX-US team for their contributions to the campaign data. Furthermore, we acknowledge the European Space Agency for making the campaign dataset available. We thank Weiwei Liu for the instruction of the DART model. We would like to thank the academic editor and the anonymous editors and reviewers for their constructive and detailed comments for improving the current manuscript.

**Conflicts of Interest:** The authors declare no conflict of interest.

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
