# Peer review of "Evaluation of the FluorWPS Model and Study of the Parameter Sensitivity for Simulating Solar-Induced Chlorophyll Fluorescence"

_remotesensing, doi:10.3390/rs13061091_

Round 1
Reviewer 1 Report
1- The proposed FluorWPS seems to be interesting.
2- I wish to congratulate authors for the development of this interesting project.
Author Response
Thank you for your comments concerning our manuscript.
Reviewer 2 Report
This paper mainly evaluated the FluorWPS model with airborne data and conducted the sensitivity analysis of SIF over a deciduous forest. This topic is of interest to the remote sensing and SIF community. However, at the current stage, the evaluation section is not very comprehensive, and I would recommend a major revision. Detailed comments are as follows.
- Only one measurement in Fig. 8 in this paper is about the direct validation of the FluorWPS model, while all the other figures are about either the data description or the sensitivity analysis. This is not very comprehensive as a model evaluation paper. It is recommended to evaluate multi-angular SIF observations and across multiple pixels with a few scatter plots if there are available datasets. One measurement is not sufficient to evaluate the accuracy of the FluorWPS model.
- The spatial representativeness of the leaf and soil spectra measurements may need to be clearly addressed to the readers. If this area is horizontal or vertical heterogenous on leaf spectra, whether the leaf and soil samples are sufficient to represent the whole region may need to be evaluated.
- In addition to the direct validation of the FluorWPS model by the airborne data, the readers would be interested in the discrepancy among various ray-tracing based 3D SIF models. It is recommended to compare the FluorWPS model with DART and FluorFLIGHT to show their potential discrepancies.
- Fig. 6: The measured tree leaf reflectance of the dashed line should be transmittance?
- There are five “leaf angel distribution” in this paper, which should be “leaf angle distribution”.
- The title and the abstract are very long with too many details. It is suggested to keep the title and the abstract concise with the main message.
- It is recommended to share the FluorWPS model and the corresponding validation to the public, such as at GitHub. The more users, the larger impact, such as the SCOPE and DART models.
- Fig. 14: The meaning of the “total SIF” here and in other figures may need to be clarified.
Reviewer 3 Report
Manuscript ID: remotesensing-1081642
Manuscript title: Evaluation of the FluorWPS model with airborne data and study of parameter sensitivity for simulating solar-induced chlorophyll fluorescence in a deciduous forest
Authors: Chiming Tong, Yunfei Bao, Feng Zhao, Chongrui Fan, Zhenjiang Li, and Qiaolin Huang
The manuscript by Tong and co-authors aims to validate the capability of the FluorWPS model to simulate hyperspectral radiance received by an airborne sensor and to identify the effects of canopy structure and sun-target-view geometry on sun-induced fluorescence (SIF) in deciduous forests. SIF receives increasing attention due to its close link to plant photosynthesis and consequently its great application potential. The topic of the manuscript fits well with the scope of the Remote Sensing journal.
In general, it is crucial to understand all biophysical factors influencing SIF yield and thus to understand and correctly interpret the remotely sensed physiological processes associated with carbon uptake (carbon cycle). From that point of the view, the manuscript deals with an important and valuable topic.
I have, however, found some peculiarities that have to be addressed before the acceptance of the manuscript.
According to my opinion, the experiment and flight campaigns seem to be well planned and executed. Experimental set up as well as data acquisition and processing are sufficiently described. This is not true for the determination of leaf area index (LAI). Since the LAI is crucial for the content of the manuscript and forms one of the conclusions (agreement of the retrieved and measured LAI value of the reconstructed scene), it is necessary to describe the field measurement of LAI in much more detail, not only one sentence (line 161). What were the radiation conditions during the measurement? how many points were measured in total? How dense was the measuring network? What was the range of LAI values found?
The second major comment relates to SIF measurement. It has to be described and explained much more carefully. The most frequently used methods to estimate sun-induced fluorescence rely on Fraunhofer Line discrimination of the oxygen absorption bands near 687 nm (O2B) and 760 nm (O2A) using infilling methods of the spectra. However, it seems like the whole spectrum ranging from 670 to 780 nm was measured (line 206). Failure to respect Fraunhofer Lines significantly reduces the applicability of the obtained results for the determination of SIF and carbon uptake from satellite images. The presented SIF signal thus contains both chlorophyll fluorescence and radiation reflected from the leaves.
At third, the study presents only single-time dataset (September 30). Moreover, this date represents a quite late autumn connected with the decrease in Chlorophyll content and the leaf senescence in general. Therefore, the presented data set may not be universally valid for the whole vegetation season. This is related to a well-known variation in both peaks at 685 nm and 740 nm during the season with variations in chlorophyll content due to overlapped spectra of fluorescent emission and chlorophyll absorption. Such a limit of the presented study has to be, at least, discussed. Also, the effects of chlorophyll and carotenoid contents have to be carefully discussed in the revised version of the manuscript.
Other specific comments:
- Introduction: Fraunhofer lines and the effect of chlorophyll and carotenoid contents have to be mentioned when introducing the concept of the sun-induced fluorescence.
- Line 57: PAR does not excite fluorescence photons; this is not the correct statement from the physical point of the view. Please, re-word.
- Line 129: “hm2” ?? Do you mean hectares (ha)?
- Line 130 and 140: The information about the forest/canopy from 2006 has to be upgraded. The spectral measurements were done in 2013. Six-seven years is a long period in forest development. Moreover, please, give the forest age.
- Figure 4 was not displayed correctly (at least in my printed version of the manuscript).
- Table 2 and lines 283-285: Please, explain PSI and PSII abbreviations.
- Equation 2: It is not entirely clear (or not explained) what SIFmax and SIFmin exactly mean. How were these parameters determined/calculated?
- Line 355: nm-1 instead of um-1
- Figure 14: Delete “70” placed between “0” and “30” (in both a and b figures).
Round 2
Reviewer 2 Report
The revised version of manuscript has been improved especially after adding the comparison with DART. Most of my concerns have been addressed, and this time I recommend a minor revision.
- The meaning of “total SIF” needs to be clarified as early as possible even in the abstract, because it is a little different from the general understanding in the SIF community. Usually we mean “the total SIF emitted by all leaves” by “total SIF”, not “the emission part comes from leaves and the scattering part comes from leaves and soil” as in this paper. In the abstract, maybe you can just use “SIF”, because usually we do not separate the directly emitted SIF and scattered SIF in terms of “observed SIF”.
- In Fig.9, there is an obvious overestimation of FluorWPS at the peak of red SIF when compared to DART. This may need a deeper exploration. From the paper, both FluorWPS and DART use the Fluspect model at the leaf scale. So the discrepancy might come from the upscaling process from leaf to canopy scale. The emission and scattered terms may need to be evaluated.
- In Table 1, as you already have the azimuth angle, I suppose the view zenith angle should be 0~70 instead of -70~70? Besides, fqe is PAR dependent for Photosystem II. In the SCOPE model, FluorWPS model and DART model, the fqe of sunlit/shaded leaves can be different according to the incoming PAR for each leaf. Whether the process of PAR-dependent fqe in the FluorWPS model and DART model is the same or not needs to be clarified.
Reviewer 3 Report
Manuscript ID: remotesensing-1081642 – revised version
Manuscript title: Evaluation of the FluorWPS model with airborne data and study of parameter sensitivity for simulating solar-induced chlorophyll fluorescence in a deciduous forest
Authors: Chiming Tong, Yunfei Bao, Feng Zhao, Chongrui Fan, Zhenjiang Li, and Qiaolin Huang
I have found the revised version of the manuscript improved. I am satisfied with how the authors responded to my critical comments. However, the manuscript requires careful proof-reading and Editorial work. There are still some inconsistencies and parts where the sentences have to be re-worded and/or better specified. Here are several examples. I recommend minor revisions.
Line 53: delete „and“ at the end of the line.
Line 59-60: The sentence „Photons in the absorbed photosynthetic active radiation (APAR) range hit the leaf then 59 excite fluorescence photons.“ does not sound logic. Should be better elaborated.
Line 133: give here the main dominant deciduous species.
Lines 134-135: It is surprising how homogenous the canopy height is for trees of 3 to 46 year old.
Line 139: Why ROI abbreviation is introduced here? I am not aware that it is used later.
Lines 160-161: „The airborne measurements were affected significantly by the atmospheric and lighting conditions, …“ It is not clear what kind of atmospheric conditions and what kind of light conditions do the authors have in mind. Should be specified.
Line 369: I am puzzled by the value of LAI. It is 3.0 m m-2 in Table 1, while it is 4.6 m m-2 on line 370.
Line 387-392: see the following lines – „For the DUAL sensor, the comparisons showed that root mean square error (RMSE) and the coefficient of determination (R2) between the measured and simulated radiance spectra in the range of 400–800 nm with a step of 3 nm were 1.85 mW·m-2·sr-1·nm-1 0.998, respectively. And for the FLUO sensor, the RMSE and R2 between the simulated and measured radiance spectra in the range of 670–782.35 nm with a step of 0.25 nm were 9.12 mW·m-2·sr-1·nm-1 and 0926, respectively.“ This part has to be better elaborated. The text is not entirely clear, there are also some peculiarities and missing words/decimal point etc.
Author Response
Please see the attachment.

This manuscript is a resubmission of an earlier submission. The following is a list of the peer review reports and author responses from that submission.